# SEPSI: A Secure and Efficient Privacy-Preserving Set Intersection with Identity Authentication in IoT

**Bai Liu** [1,*] , **Xiangyi Zhang** [1] , **Runhua Shi** [1] , **Mingwu Zhang** [1,*] and **Guoxing Zhang** [2]

1 The School of Computer Science, Hubei University of Technology, Wuhan 430068, China; xiangyizhang@foxmail.com (X.Z.); hfsrh@sina.com (R.S.)
2 School of Management, Lanzhou University, Lanzhou 730000, China; guoxingzh@lzu.edu.cn
* Correspondence: liubai@hbut.edu.cn (B.L.); csmwzhang@gmail.com (M.Z.)

**Abstract:** The rapid development of the Internet of Things (IoT), big data and artificial intelligence (AI) technology has brought extensive IoT services to entities. However, most IoT services carry the risk of leaking privacy. Privacy-preserving set intersection in IoT is used for a wide range of basic services, and its privacy protection issues have received widespread attention. The traditional candidate protocols to solve the privacy-preserving set intersection are classical encryption protocols based on computational difficulty. With the emergence of quantum computing, some advanced quantum algorithms may undermine the security and reliability of traditional protocols. Therefore, it is important to design more secure privacy-preserving set intersection protocols. In addition, identity information is also very important compared to data security. To this end, we propose a quantum privacy-preserving set intersection protocol for IoT scenarios, which has higher security and linear communication efficiency. This protocol can protect identity anonymity while protecting private data.

**Keywords:** private set intersection; quantum authentication; oblivious quantum key distribution; Internet of Things

## 1. Introduction

In the Internet of Things (IoT), many devices are connected to exchange data through the internet [1,2]. The core components of IoT are smart devices, the internet and connectivity, where IoT devices collect information about personal behavior. In recent years, the development of IoT has brought about many practical scenarios, such as the Internet of Medical Things (IoMT) [3], smart cities [4], and smart homes [5]. IoT services bring great convenience to human life.

As a basic service, privacy-preserving set intersection (PSI) in IoT is widely used in various practical environments. For example, in IoMT, hospitals cannot share electronic medical records while protecting patient privacy. Patients with similar symptoms also cannot exchange and share medical information. Therefore, there exists the phenomenon of information islands in IoMT. In this regard, personal health information (PHI) can be securely shared through profile matching [6] based on PSI. In a cloud environment, Abadi et al. [7] proposed an efficient delegated privacy set intersection scheme on outsourced private datasets. In addition, private graph intersection operation also plays an important role in social networks. Zuo et al. [8] proposed an efficient and privacy-preserving verifiable graph intersection scheme using cryptographic accumulators in social networks.

Because of its importance and wide applicability, many privacy-preserving set intersection (PSI) protocols have been proposed. In 2004, Friedman et al. [9] proposed the first PSI protocol, where a set can be used with homomorphic encryption to ensure secure computation. In 2019, Le et al. [10] proposed a PSI protocol based on secret sharing, which removes the trusted third party of the protocol [11]. Kolesnikov et al. [12] proposed a

new PSI protocol, which improved the communication efficiency of the protocol [13] by 2.9–3.3 times. In 2020, Chase et al. [14] proposed a novel lightweight multi-point oblivious pseudorandom function protocol based on oblivious OT extension and utilized it to construct a PSI scheme. In 2021, Badrina Rayanan et al. [15] proposed an updated privacy set intersection protocol, which allows two parties that have constantly updated sets to calculate their privacy set intersections.

However, most existing PSI protocols are based on difficulty assumptions, which are vulnerable to attacks by quantum technology. As a consequence, classical PSI protocols may not have long-term security and the design of quantum-resistant PSI protocols becomes a research hot spot. In addition, quantum cryptography [16,17] has emerged, which can guarantee information-theoretic security.

In this article, we propose a general system model of privacy-preserving set intersection in IoT, which is aided with edge computation (ED). Then, we present a quantum protocol for a private-preserving set intersection with identity authentication. A novel quantum PSI in IoT is designed with the help of obvious quantum key distribution, quantum authentication and count Bloom filter.

Our contributions, in this paper, are summarized as follows:

- We propose a general system model aided with ED of PSI, which is suitable for IoT applications.
- we present a novel quantum updatable PSI protocol in IoT, which can be roughly divided into three phases: key generation, encryption and decryption.
- We analyze security and communication efficiency of the protocol. The protocol has efficient communication efficiency, i.e., linear communication complexity $O(\tau)(\tau \ll N)$ qubits, where N is the size of the universal set. The proposed protocol has higher security. The protocol also provides identity authentication to protect identity information and to maintain the integrity of the transmitted information.

The remainder of this article is organized as follows. In Section 2, we introduce the related works of a privacy-preserving set intersection in a quantum setting. Then, we describe our system model, security model and design goals in Section 3. In Section 4, we present our quantum PSI protocol, followed by security analysis and performance evaluation in Section 5. Then, we have some discussions in Section 6. Finally, we draw our conclusions in Section 7.

## 2. Related Works

### 2.1. Quantum PSI Protocol

In 2015, Shi et al. [18] first proposed a cheat-sensitive quantum PSI protocol using phase-encoded private query. Then, Cheng et al. [19] presented a new quantum PSI protocol, which is cryptanalysis and an improvement of the protocol [18]. Cheng's protocol shows that the protocol [18] is not as efficient as claimed because the communication complexity should be $O(nlogN)$ instead of $O(n)$. Later, Maitra [20] presented a fair quantum PSI protocol based on a set membership decision protocol [21]. However, these protocols need complicated oracle operators and multi-particle entangled states. Subsequently, in order to enhance the realizability, Kumar [22] introduced a feasible quantum private set intersection protocol with single photons using the flexible oblivious quantum key distribution (OQKD) [23]. Based on the quantum PSI protocol [22], Debnath et al. [24] presented an efficient quantum PSI protocol, which reduced communication complexity. However, a multi feasible OQKD protocol [25] was broken by the protocol [26] using the man-in-the-middle attack. Therefore, the security of protocols [22,24] may not be guaranteed.

### 2.2. Oblivious Quantum Key Distribution

In 2011, Jakobi et al. [27] proposed a practical oblivious quantum key distribution (OQKD) protocol, which guaranteed better efficiency and feasibility of a private quantum query. The oblivious key can be distributed between two parties by using SARG04 QKD [28],

where the sender knows the whole key while the receiver only knows a single or a few bits of the key. The main process of OQKD can be briefly described as follows:

The sender, i.e., Alice, generates a long quantum sequence including states $|\uparrow\rangle, |\downarrow\rangle$, $|\leftarrow\rangle, |\rightarrow\rangle$, where two quantum states carry a bit of classical information, e.g., $\{|\downarrow\rangle, |\uparrow\rangle\}$ represent the bit 0 and $\{|\leftarrow\rangle, |\rightarrow\rangle\}$ denote the bit 1. Then, Alice sends the quantum sequence to the receiver. After receiving it, the receiver, i.e., Bob, measures each qubit randomly in $\leftrightarrow$ basis or $\updownarrow$ basis.

Then, Bob announces that he successfully measured the positions of the qubits and discards the missed or undetected qubits. For each qubit that Bob successfully measured, Alice announces a pair of verification qubits to verify the correctness of Bob's measured results. Due to the uncertainty of measurements in quantum mechanics, Bob only obtains partial values that match a pair of qubits published by Alice. In other words, Bob can only obtain partially correct values of the key. In order to reduce Bob's information on the raw key, two parties cut the raw key into multiple substrings of length N and added these strings bitwise to obtain the final key with length N.

Then, Gao et al. [23] proposed a variant OQKD protocol in which a variable angle $\theta$ was introduced in the protocol [27]. That is, they use four generalized states $\{|0\rangle, |1\rangle, |0'\rangle, |1'\rangle\}$, where $|0'\rangle = cos\theta|0\rangle + sin\theta|1\rangle$ and $|1'\rangle = cos\theta|0\rangle - sin\theta|1\rangle$.

Later, Xiao et al. [29] integrated an identity authentication mechanism into the OQDK protocol [27] to present a new OQKD protocol that can implement mutual identity authentication to resist malicious adversary attacks. First, two parties register with a trusted third party (Certificate Authority, CA) to obtain their respective identity information, i.e., Alice's identity string $ID_C$ and Bob's identity string $ID_S$. Then, Alice sends the qubits used as the original key (QOK) along with the qubits for authentication (QA) to CA. All qubits need to be forwarded by the CA to Bob, where QA are modified by the CA based on the identity strings of both parties. Both parties can authenticate with QA to obtain a key $K$ that can be used for subsequent anonymous authentication. Another difference with the OQDK protocol [27] is that instead of directly disclosing the quantum bit pairs used to verify Bob's measurement results, Alice encrypts them with the key $K$ and sends them to Bob. The system model is shown in Figure 1.

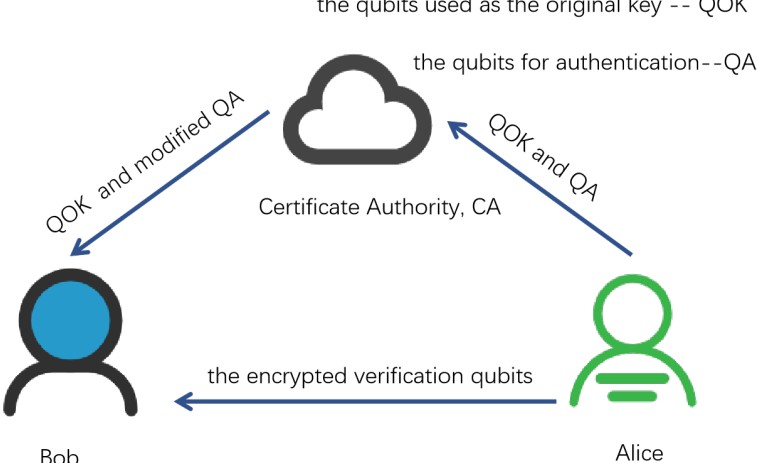

**Figure 1.** System model of the OQKD protocol [29].

### 2.3. Quantum Authentication

Quantum message authentication is an important research direction in quantum cryptography and is divided into two parts: authentication of classical information [30] and authentication of quantum information [31]. Curty et al. [30] proposed the first protocol for classical information by quantum entangled states. Subsequently, Xi et al. [32] proposed a quantum authentication scheme that required only single photons. This protocol assumes that two parties pre-share a classical key and a pair of quantum operators. Then, the sender

converts a classical message into quantum bits and transmits these qubits to the receiver through the quantum channel. Finally, the receiver verifies the authenticity of the qubits.

The main process of the protocol [32] is as follows:

Suppose that Alice has a classical set $\{m_1, m_2, ...m_n\}$, where $m_i \in \{0,1\}$ and $i \in \{1,2,...,n\}$. Two parties, Alice and Bob, share a secret key $\{s_1, s_2, ...s_{n+1}\}$ in advance, where $s_i \in \{0,1\}$ and $i \in \{1,2,...,n+1\}$. Then, two parties also pre-share two publicly quantum unitary operations, $U_0$ and $U_1$, which should satisfy the following conditions:

1. $U_0|v\rangle\langle v|U_0^+ + U_1|v\rangle\langle v|U_1^+ \neq 0$.
2. There is no a unitary operation $U_e$ to make $\langle v|U_i^+ U_e U_i|v\rangle = 0$, where $i \in \{0,1\}$.
3. $\langle v|U_0^+ U_1|v\rangle \neq 0$.

where $|v\rangle$ is an arbitrary qubit.

Two parties select two pairs of arbitrary quantum states, i.e., $|\varphi_0\rangle, |\varphi_1\rangle$, and $|\psi_0\rangle, |\psi_1\rangle$, where $\langle \varphi_0|\varphi_1\rangle = 0$ and $\langle \psi_0|\psi_1\rangle = 0$. As shown in Tables 1 and 2, Alice generates a pair of quantum states $\{|a_i\rangle|t_i\rangle\}$, where the first qubit represents the quantization of $m_i$ and the second qubit implies the relevant label of $m_i$. Alice transforms classical information $\{m_1, m_2, ...m_n\}$ to obtain a quantum sequence $\{|a_1\rangle, |t_1\rangle, |a_2\rangle, |t_2\rangle, ...|a_n\rangle, |t_n\rangle\}$ by the method in Tables 1 and 2, then sends the quantum sequence to Bob.

After receiving the quantum sequence, Bob selects suitable measurement bases by the method in Table 3, then measures the quantum sequence $\{|a_1\rangle, |t_1\rangle, |a_2\rangle, |t_2\rangle, ...|a_n\rangle, |t_n\rangle\}$. If each quantum pair satisfies the equation $|t_i\rangle_m = U_{s_{i+1}}|a_i\rangle_m$, where $|t_i\rangle_m$ and $|a_i\rangle_m$ are measurement results of $|t_i\rangle$ and $|a_i\rangle$, respectively, the quantum sequence passes the verification of Bob.

**Table 1.** The value of $|a_i\rangle$.

| $s_i/m_i$ [1] | 0 | 1 |
|---|---|---|
| 0 | $\|a_i\rangle = \|\varphi_0\rangle$ | $\|a_i\rangle = \|\varphi_1\rangle$ |
| 1 | $\|a_i\rangle = \|\psi_0\rangle$ | $\|a_i\rangle = \psi_1\rangle$ |

[1] The row represents the value of $m_i$, while the column represents the value of $s_i$.

**Table 2.** The value of $|t_i\rangle$.

| $s_{i+1}$ | $\|t_i\rangle$ |
|---|---|
| 0 | $U_0\|a_i\rangle$ |
| 1 | $U_1\|a_i\rangle$ |

**Table 3.** Measurement basis of $|t_i\rangle$.

| $s_{i+1}/s_i$ [1] | 0 | 1 |
|---|---|---|
| 0 | $\{U_0\|\varphi_0\rangle, U_0\|\varphi_1\rangle\}$ | $\{U_1\|\varphi_0\rangle, U_1\|\varphi_1\rangle\}$ |
| 1 | $\{U_0\|\psi_0\rangle, U_0\|\psi_1\rangle\}$ | $\{U_1\|\psi_0\rangle, U_1\|\psi_1\rangle\}$ |

[1] The row represents the value of $s_i$, while the column represents the value of $s_{i+1}$.

### 2.4. Count Bloom Filter

A Bloom filter is an efficient data structure that is mainly used to determine or find whether an element exists in a set. The Bloom filter was first proposed by B.H. Bloom in 1970 [33]. Since Bloom filters do not support delete operations, it cannot be adapted to dynamic data environments. A counting Bloom filter that can support a delete operation is proposed in the protocol in [34].

Figure 2 shows the composition of a counting Bloom filter. It mainly consists of two tools: an array of size $m$ and $k$ different collision-resistant hash functions $\{H_1, ..., H_k\}$, where $H_i : \{0,1\}^* \longrightarrow \{1, ..., m\}$ for $i \in \{1,2,...,k\}$. Suppose Alice has a private set $S = \{s_1, s_2, ..., s_n\}$. She wants to map all elements of $S$ into the m-size array $CBF_s$ by $k$ hash functions. Initially, Alice obtains an empty array $CBF_s$, where all elements are set

to 0. For each element $x$ of $S$, Alice uses hash functions $\{H_1, ..., H_k\}$ to obtain positions $\{H_1(x)th, ..., H_k(x)th\}$ in $CBF_s$, and adds 1 to the values in these positions.

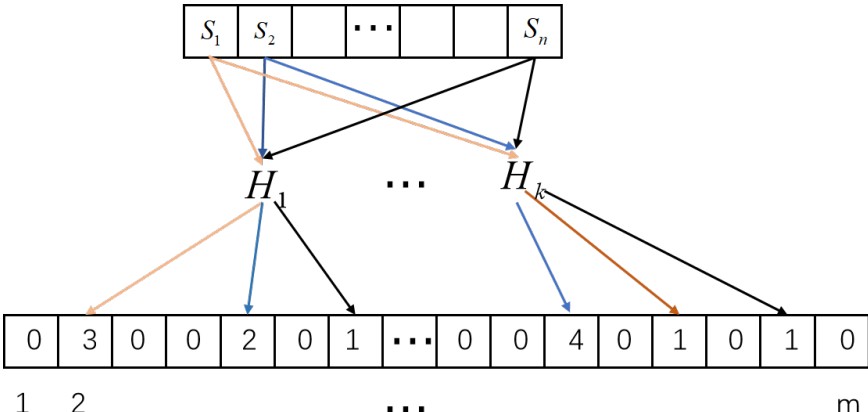

**Figure 2.** Counting bloom filter.

In general, if someone wants to insert an element into $CBF_s$, he can use hash functions to map the element to the corresponding positions in $CBF_s$ and add one to the values in these positions. In addition, if someone wants to query whether an element $x$ belongs to $S$, he only needs to map the element $x$ to the corresponding positions in $CBF_s$ by hash functions. Then, he determines whether the values in all these positions are non-zero. If there exists a position where the value is 0, then it means that the element x cannot belong to S. If Alice wants to delete an element $x$ of $S$ to $CBF_s$, she only needs to map the element $x$ to the corresponding positions in $CBF_s$ and reduces the value of all positions by one unit (value = value − 1). Please note that $x$ must belong to $S$. However, if the values in all positions are non-zero, it is possible that $x$ is not in S. That is, the count Bloom filter has false positives.

### 3. Models and Design Goal

#### 3.1. System Model

In this section, we will illustrate our design of the privacy-preserving set intersection from a system perspective. Our system model consists of five groups of entities: (1) IoT devices; (2) devices for an edge device; (3) a server provider; (4) a client and (5) a certificate authority, as shown in Figure 3.

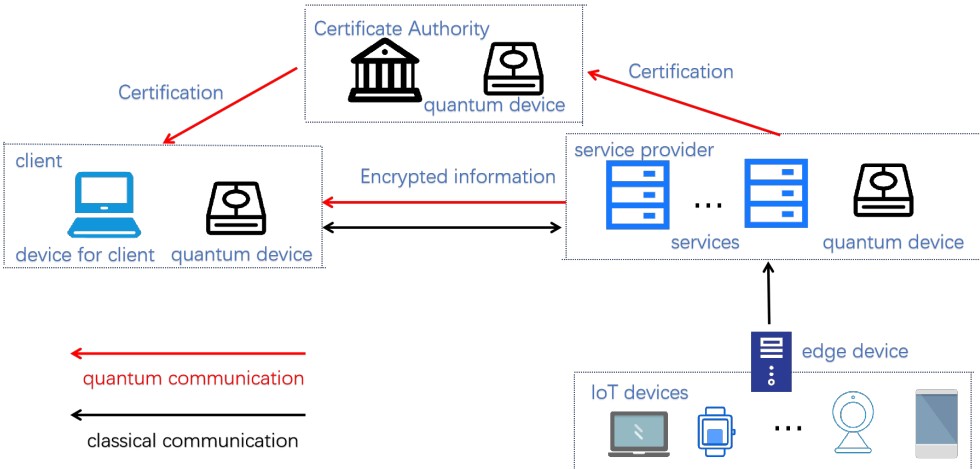

**Figure 3.** System model aided with ED of PSI in IoT scenarios.

IoT Devices: IoT devices equipped with sensing and communication capabilities are deployed in areas of interest. IoT devices generate real-time data and periodically report data to the edge device. Communication between IoT devices and edge devices is classic communication.

Edge Devices (ED): In order to improve efficient communication, an edge device is deployed at the network edge, which receives the data reported from IoT devices. After receiving data, it locally processes, aggregates, and forwards data to a service provider.

Service Provider (SP): An SP might consist of servers equipped with quantum devices. The SP directly provides the IoT services to the end client. Specifically, we take the IoMT scenario as an example to describe the privacy-preserving set intersection. In hospitals, various IoT devices monitor patients' physical health, such as physiological parameters and living habits. After IoT devices report data to an ED, ED first processes data locally. Then, the ED forwards processing results to SP through wireless communication. When physicians belonging to other hospitals want to obtain data on patients with similar diseases, SP will respond to the client according to this protocol.

Client: A client may be an end device that is equipped with quantum devices. She receives anonymous encrypted data from SP and calculates the privacy intersection of their sets.

Certificate Authority (CA): A CA is a trusted third party that generates identity information for clients and servers. CA is also equipped with quantum devices that can forward quantum states to the client. CA is only used in the basic building block of the protocol: oblivious key distribution scheme [29], which is introduced in Section 2.2.

In this paper, the quantum devices required above only need to support single-photon preparation, measurement, and simple single-bit operations. That is, the quantum device we describe is not a full-fledged quantum computer including quantum random memory but has some basic devices [35–38] and single-bit circuits that can support single-photon operations.

### 3.2. Security Model

We consider honest-but-curious parties, where adversaries may attempt to learn more information from a given protocol execution but are not able to deviate from the protocol.

**Definition 1.** *Privacy-preserving set intersection (PSI) protocol—there are two communicating parties, i.e., a client with a private set C and an SP with a private set S. After executing a PSI protocol, the client outputs the intersection of their respective private sets, i.e., $C \cap S$, but the SP obtains nothing. Furthermore, a PSI protocol should meet the following privacy requirements:*

*(1) SP Privacy: The client learns no information about the SP's private set except the intersection $C \cap S$.*

*(2) Client Privacy: SP cannot obtain any private information about the client's private set.*

Traditionally, PSI uses a static setting where computation is performed only once on both parties' input sets. We also consider that parties can periodically calculate the intersection of their private updatable sets.

In addition, we also consider external adversary attacks and authentication analysis to enhance security. That is, the protocol should also meet the following security requirement:

(3) Authentication: If the tag passes authentication, the client will continue to execute the protocol, otherwise, terminate.

Due to the focus on privacy-preserving of two parties, i.e., a client and an SP, during the interaction, we do not consider the honesty of IoT devices and EDs. That is, they faithfully report data and are not subject to attack.

### 3.3. Design Goal

The design goals are as follows.

- The proposed protocol can not only protect the private data of both parties but also protect the identity information of both parties. The protocol needs to ensure correctness without losing the ability to protect privacy. In order to enhance privacy protection, the protocol is required to protect the identity information of both parties. In addition, the protocol may be subject to external attacks with quantum devices, so it needs to have a certain resistance to external attacks.
- The proposed protocol should have efficient communication efficiency. This protocol only needs the linear communication complexity of $O(\tau)$ qubits.

## 4. Proposed Protocol

In this protocol, assume that a client has a private set $C = \{c_1, c_2, ..., c_v\}$ and an SP has a private set $S = \{s_1, s_2, ..., s_w\}$, where $w > v$. All elements of sets $C$ and $S$ lie in $Z_N$, where $Z_N = \{0, 1, 2, .., N - 1\}$.

Furthermore, SP and the client have the same count Bloom filter parameters, i.e., hash functions $\{h_1, h_2, ..., h_\lambda\}$ and the length $\tau$ of the count Bloom filter [34,39].

The protocol consists of three main parts, including key generation phase, encryption phase and decryption phase. Next, we will describe these phases. In addition, specific notations used in the following text are illustrated in Table 4.

**Table 4.** Definitions of notations.

| Notations | Definitions |
|---|---|
| $C$ | The client's private set |
| $S$ | The SP's private set |
| $\{h_1, h_2, ..., h_\lambda\}$ | The hash functions |
| $\tau$ | The length of the count Bloom filter |
| $k_B$ | The raw key distributed by the SP |
| $K$ | The message authentication key from the protocol [29] |
| $k_b$ | The intermediate key after checking the SP's honesty |
| $k_b^*, k^*$ | The final key distributed by the SP |
| $CBF$ | The SP's count Bloom filter |
| $BF$ | The variant of $CBF$ |
| $KBF$ | Encryption result of the array $BF$ by the key $k^*$ |
| $|a_i\rangle, |t_i\rangle$ | The $i$th element of the encryption result of the array $KBF$ by the key $K$ |
| $CBF_C$ | The client's count Bloom filter |
| $\{p_{1,}, p_2..., p_m\}$ | The positions index of non-zero items of $CBF_C$ |

### 4.1. Key Generation

In this section, two parties, i.e., a client and an SP, will be distributed a special asymmetric key. SP knows every bit of the key, while the client only knows partial bits of the key, where each bit that the client knows is associated with a unique element of her private set. For instance, assume that position indexes of the key bits start from 0 to $N - 1$. Suppose that Alice has a set $X = \{x_1, x_2..., x_n\}$, where $x_i \in \{0, 1, ,...N - 1\}$ and $n < N$. Then, Alice only knows the $x_1 th, x_2 th, ...$and $x_n th$ bits of the key.

Step 1: The client and SP invoke Xiao's Oblivious Quantum Key Distribution (OQKD) protocol [29] to share a random secret $(\tau + q)$-bit key $k_B$. SP knows the whole key $k_B$, and the client only knows $m + q$ bits of key $k_B$ (note that m is the number of non-zero items in the client's array $CBF_C$ during decryption phase, $\tau$ is the size of SP's array BF in encryption phase and q is a security parameter).

Furthermore, as for reference [29], we can also obtain the $\tau + 1$ bits message authentication key $K$, which only are known by the SP and client.

Step 2: Then, the client randomly chooses q bits of the key to check whether SP is honest. That is, she requests SP to announce the values of these checked bits. If these values published by SP do not entirely match those that she has deciphered, it would indicate that SP is dishonest or there is an outside eavesdropper. If the client discovered a dishonest SP

or any outside eavesdropping, she would terminate this protocol, otherwise, continue to the next step.

Step 3: SP and the client discard q checked bits of the raw key $k_B$ and further obtain the intermediate key $k_b$ of length $\tau$. Similarly, the client only knows m bits of key $k_b$, while SP still knows all bits. Actually, the client knows not only m-bit values: $k_b(j_1), k_b(j_2), ..., k_b(j_m)$ but also their respective position indexes: $\{j_1, j_2, ..., j_m\}$, where $k_b(j_i)$ denotes the $j_i$ th bit of $k_b$. In addition, SP does not know the bits which the client knows.

Step 4: The client generates a random permutation $\pi$ of an $\tau$-element sequence by position index set $\{j_1, j_2, ..., j_m\}$ and non-zero items' position index set $\{p_1, p_2..., p_m\}$ of the count Bloom filter $CBF_C$, which must meet the following condition

$$\{k_b(j_1), \ldots, k_b(j_m)\} = \{k_b^*(p_1),, \ldots, k_b^*(p_m)\} \tag{1}$$

where $k_b^*$ is a new sequence after applying the permutation $\pi$ to $\tau$-element sequence $k_b$, i.e., $k_b^* = \pi(k_b)$. Then the client announces the permutation $\pi$ to SP.

Step 5: SP obtains the final key $k_b^* = \pi(k_b)$ from key $k_b$ by permutation $\pi$. Obviously, the client only knows partial bits: $k_b^*(p_1), k_b^*(p_2), \ldots, k_b^*(p_m)$, where $k_b^*(p_i)$ denotes the $p_i th$ bit of $k_b^*$ for $i = \{1, 2, ..., m\}$. However, SP does not know any secret information about position index set $\{p_1, p_2..., p_m\}$ without $\{j_1, j_2, ..., j_m\}$.

Here, we give a simple example to illustrate how to generate an oblivious key between the client and SP, as shown in Figure 4. The client and SP share the length $\tau = 14$ of the count Bloom filter. The client has position indexes, $\{p_1 = 4, p_2 = 7, p_3 = 8, p_4 = 14\}$, of non-zero items in the count Bloom filter, and thus finally, she only knows $k_b^*(4), k_b^*(7), k_b^*(8)$ and $k_b^*(14)$, while SP knows all bits of $k_b^*$. The elements of Figure 4 with blue background are the checked qubits, such as $k_B(11)$ and $k_B(15)$. The elements with black slashes are the checked qubits that have been discarded, such as $k_b(15)$ and $k_b(16)$.

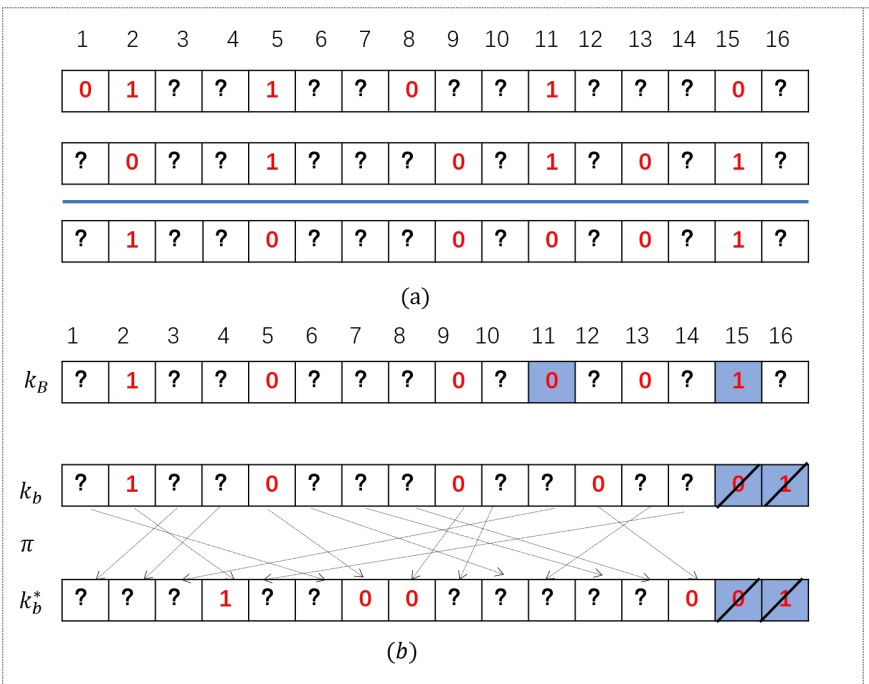

**Figure 4.** Illustration of generating the key. (**a**) How to reduce the client's information in the key. (**b**) How to obtain the final key $k_b^*$ from the raw key $k_B$.

### 4.2. Encryption

Suppose that SP has a private set $S = \{s_1, s_2, ..., s_w\}$, where every element lies in $Z_N$. She employs $\lambda$ independent collision resistant hash functions $\{h_1, h_2, ..., h_\lambda\}$.

Step 6: In this step, SP utilizes Algorithm 1 to generate an array of $\tau$ elements. First, SP maps the private set $S = \{s_1, s_2, ..., s_w\}$ to the counting Bloom filter $CBF = \{CBF_1, CBF_2, ..., CBF_\tau\}$ through hash functions $\{h_1, h_2, ..., h_\lambda\}$. Then, SP selects an array $BF = \{BF_1, BF_2, ..., BF_\tau\}$, where all elements initialize to 0. All elements of corresponding positions in $BF$ are set to 1, according to non-zero items in $CBF$. SP has position indexes $\{q_1, q_2, ..., q_l\}$ of non-zero items in the array $BF$. The construction process is shown in Figure 5.

Furthermore, SP's database is constantly changing in the actual environment. Therefore, SP synchronously modifies the local counting Bloom filter through Algorithms 2 and 3.

---

**Algorithm 1** Generating an array of $\tau$ elements

---

**Require:** $\{s_1, s_2, ..., s_w\}$.
**Ensure:** $BF \in \{0, 1\}^\tau$.
1: **for** $i = 1$ to $\tau$ **do**
2:     $CBF[i] = 0$
3:     $BF[i] = 0$
4: **end for**
5: // All $\tau$ elements in $CBF$ and $BF$ are set to 0 initially.
6: **for** $i = 1$ to $w$ **do**
7:     **for** $j = 1$ to $\lambda$ **do**
8:         $CBF[h_j(s_i)] = CBF[h_j(s_i)] + 1$;
9:     **end for**
10: **end for**
11: // That is, for each element $s_i$ of the private set S, the $h_1(s_i)th, h_2(s_i)th, ...,$and $h_\lambda s_i th$ the elements of $CBF$ all plus 1.
12: **for** $i = 1$ to $\tau$ **do**
13:     **if** CBF[i] > 0 **then**
14:         BF[i] = 1;
15:     **end if**
16: **end for**
17: // That is, for each element $s_i$ of the private set S, the $h_1(s_i)th, h_2(s_i)th, ...,$and $h_\lambda(s_i)th$ the elements of $BF$ all set 1.

---

**Algorithm 2** Adding an element to count Bloom filter

---

**Require:** $x$.
**Ensure:** $BF$ and $CBF$, where $CBF = CBF \cup x$.
1: Execute Algorithm 1 to generate CBF and BF
2: **for** $i = 1$ to $\lambda$ **do**
3:     $CBF[h_i(x)] = CBF[h_i(x)] + 1$;
4:     **if** BF[i] = 0 **then**
5:         BF[i] = 1;
6:     **end if**
7: **end for**

---

---

**Algorithm 3** Deleting an existing element from count Bloom filter

---

**Require:** $x$;
**Ensure:** $BF$ and $CBF$, where $CBF = CBF - x$;
1: Execute Algorithm 1 to generate CBF and BF
2: **for** $i = 1$ to $k$ **do**
3:      CBF[i] = CBF[i]-1;
4:      **if** CBF[i] = 0 **then**
5:          BF[i] = 0 ;
6:      **end if**
7: **end for**
8: //Please note that it must guarantee that the element indeed belongs to the set associated with count Bloom filter before deleting it.

---

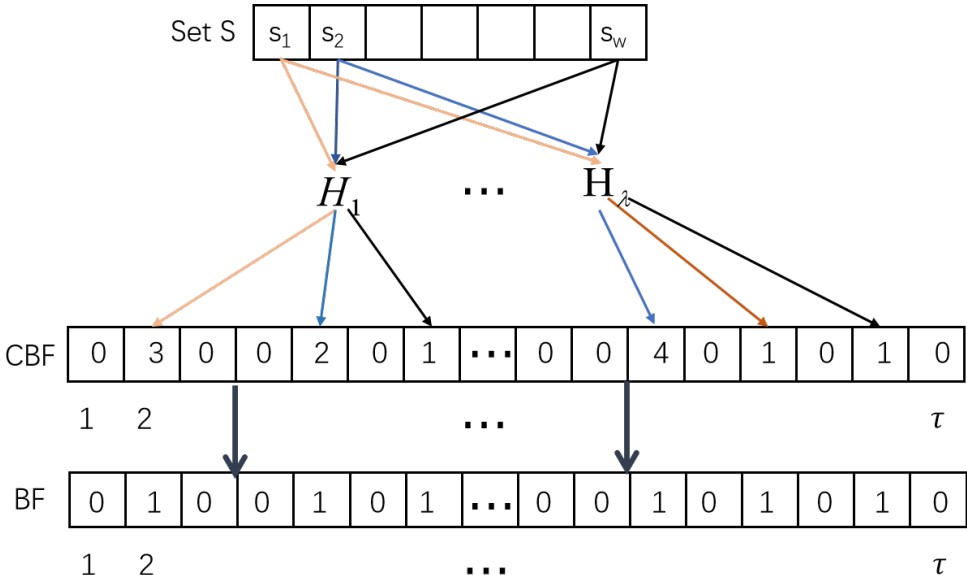

**Figure 5.** The process of transforming data.

Step 7: After obtaining the array BF, SP encrypts it with the key $k^* (k^* = k_b^*)$ to obtain

$$
\begin{aligned}
KBF &= k^* \oplus BF \\
&= \{k_1^* \oplus BF[1], k_2^* \oplus BF[2], ..., \{k_n^* \oplus BF[\tau]\} \\
&= \{KBF_1, ...KBF_\tau\}.
\end{aligned}
\tag{2}
$$

Then, as for reference [32], the client and SP publicly select two unitary quantum operations $U_0, U_1$, which should satisfy the conditions of Section 2.3.

According to the key $K$ and operations $U_0, U_1$, SP transforms $KBF\}$ into $\tau$ pairs of qubits $\{|a_1\rangle, |t_1\rangle, |a_2\rangle, |t_2\rangle, ..., |a_\tau\rangle, |t_\tau\rangle\}$, where each item $KBF_j$ is associated with a pair of qubits $|a_j\rangle, |t_j\rangle$. First qubit $|a_j\rangle$ is the quantization of $KBF_j$ and the second $|t_j\rangle$ is the tag of $KBF_j$. Finally, SP sends this quantum sequence to the client.

*4.3. Decryption*

Suppose that a client has a private set $C = \{c_1, c_2, ..., c_v\}$, where every element lies in $Z_N$. He also employs $\lambda$ independent collision resistant hash functions $\{h_1, h_2, ..., h_\lambda\}$.

Step 8: The client also generates a count Bloom filter $CBF_C$ of $\tau$ size and can obtain position indexes of non-zero items in $CBF_C$, i.e., $\{p_1, p_2, ..., p_m\}$.

Furthermore, the client's database is also constantly changing in actual environments. Therefore, the client synchronously modifies the local counting Bloom filter through

Algorithms 2 and 3. However, different from SP, the client does not need to generate the array $BF_C$ that is similar to $BF$.

Step 9: After receiving the quantum sequence from SP, the client verifies each pair of qubits. As previously introduced in Section 2.3, if the client finds that equation $|t_i\rangle_m = U_{K_{i+1}}|a_i\rangle_m$ holds, where $i \in \{1, 2, ..., \tau\}$, the verification will succeed, otherwise, it will fail. $|t_i\rangle_m$ and $|a_i\rangle_m$ are measurement results of $|t_i\rangle$ and $|a_i\rangle$, respectively. If the client discovered a dishonest SP or any outside eavesdropping, she would terminate this protocol, otherwise, continue to the next step. After successful authentication, the client obtains a correct encrypted array $KBF = \{KBF_1, ..., KBF_\tau\}$. Then the client decrypts $KBF$ to obtain decrypted values of partial position indexes $\{p_1, p_2, ..., p_m\}$ in $KBF$ by $k^*$, where the client only knows m bits of $k^*$. Furthermore, the decryption of array $KBF$ is also reflected in Algorithm 4.

Finally, the client continues to execute Algorithm 4 to obtain the desired private set intersection $C \cap S$.

---

**Algorithm 4** Obtaining the set intersection

---

**Require:** $C = \{c_1, c_2, ..., c_v\}, KBF, \{p_1, p_2, ..., p_m\}, k^*$;
**Ensure:** $\chi \in \{0, 1..., N-1\}^\tau$, where $\chi = C \cap S$;
 1: **for** $i = 1$ to $\tau$ **do**
 2:     PBF[i] = 0;
 3:     $\chi$[i] = 0;
 4: **end for**
 5: **for** $i = p_1$ to $p_m$ **do**
 6:     PBF[i] = KBF[i] $\oplus k^*$[i];
 7: **end for**
 8: //Initialization and setting values;
 9: z = 0;
10: **for** $i = 1$ to $v$ **do**
11:     **for** $j = 1$ to $\lambda$ **do**
12:         **if** $PBF[h_j(c[i])] = 0$ **then**
13:             Break;
14:         **end if**
15:     **end for**
16:     $\chi$[++z] = c[i];
17: **end for**
18: //Testing membership tests

---

## 5. Security Analysis and Performance Evaluation

In this section, we mainly analyze the security and performance evaluation of this protocol. In the above definition 1, PSI protocol satisfies the following three security properties:

1. Correctness: After executing the protocol, the client should obtain the correct set intersection $(C \cap S)$.

2. SP Privacy: The client learns no information about SP's set except $C \cap S$.

3. Client Privacy: SP cannot obtain any private information about the client's set.

Next, we specifically analyze three properties of this protocol.

### 5.1. Correctness

As we know, the client has a private set $C = \{c_1, c_2, ..., c_v\}$ and SP has a private set $S = \{s_1, s_2, ..., s_w\}$, where $w > v$. All elements of sets, i.e., C and S, lie in $Z_N$, where $Z_N = \{0, 1, 2, .., N-1\}$.

Furthermore, SP and the client have same count Bloom filter parameters: hash functions $\{h_1, h_2, ..., h_\lambda\}$ and the size $\tau$ of the count Bloom filter. Then, SP has position indexes

$\{q_1, q_2, ..., q_l\}$ of non-zero items in $BF$. The client also has position indexes $\{p_1, p_2, ..., p_m\}$ of non-zero items in the count Bloom filter $CBF_C$. Then, we will obtain

$$
\begin{aligned}
i \in S \cap C &\Longleftrightarrow i \in S \wedge i \in C \\
&\Longrightarrow BF[j] \neq 0 \wedge CBF_C[j] \neq 0 \\
&\wedge j \in \{h_1(i), h_2(i), ..., h_\lambda(i)\} \\
&\quad \text{(by hash functions } \{h_1, h_2, ..., h_\lambda\}) \\
&\Longrightarrow BF[j] \neq 0 \wedge j \in \{p_1, p_2, ..., p_m\} \\
&\Longrightarrow BF[j] \wedge j \in \{q_1, q_2, ..., q_l\} \wedge j \in \{p_1, ..., p_m\} \\
&\Longrightarrow BF[j] \wedge j \in \{q_1, q_2, ..., q_l\} \cap \{p_1, p_2, ..., p_m\} \\
&\Longrightarrow KBF[j] \wedge j \in \{q_1, q_2, ..., q_l\} \cap \{p_1, p_2, ..., p_m\} \\
&\quad \text{(by Equations (2))} \\
&\Longrightarrow PBF[j] \wedge j \in \{q_1, q_2, ..., q_l\} \cap \{p_1, p_2, ..., p_m\} \\
&\quad \text{(by step 1} \curvearrowright 7 \text{ of Algorithm 4)} \\
&\Longrightarrow i \in \chi \Longleftrightarrow i \in S \cap C \\
&\quad \text{(by step 10} \curvearrowright 17 \text{ Algorithm 4)}
\end{aligned}
$$

Therefore, the set of all parameters $i$ satisfying condition $i \in \chi$ is equal to the intersection of their respective private sets, i.e., $C \cap S$. Thus, the proposed protocol is correct.

Furthermore, we give an example to clearly illustrate correctness of the protocol from Figure 6. In this example, the client has a private set $C = \{25, 34, 56, 36, 57\}$ and SP has a private set $S = \{20, 34, 56, 38, 50\}$, where all elements of sets $C$ and $S$ lie in $Z_{60}$.

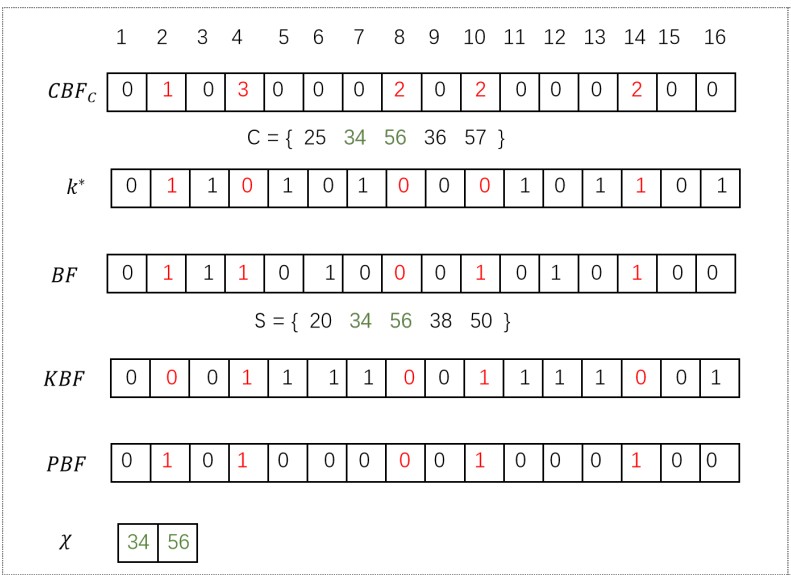

**Figure 6.** An example of privately computing $C \cap S$.

Two parties have the same count Bloom filter parameters: hash functions $h_1, h_2$ and the length of the count Bloom filter $\tau = 16$. First, SP and the client successfully construct their own count Bloom filters, i.e., $CBF$ and $CBF_C$. In addition, SP extends count Bloom filter $CBF$ to obtain an array $BF$. Then, SP has position indexes $\{2, 3, 4, 6, 10, 12, 14\}$ of non-zero items in $BF$. The client also has position indexes $\{2, 3, 4, 6, 10, 14\}$ of non-zero items in the count Bloom filter $CBF_C$.

In addition, the quantum sequence $\{a_1, t_1, a_2, t_2 ..., a_\tau, t_\tau\}$ has no influence on the correctness of the protocol. Therefore, we do not consider the quantum sequence in the following example.

After the key generation phase, SP secretly obtains the final key $k^*(k^* = k_b^*)$, where the client obtains values of position indexes of red digits in the key $k^*$. Obviously, $CBF_C[h_1(i)] \neq 0$ and $CBF_C[h_2(i)] \neq 0$ if $i \in C$. If $i \in S$, $BF[h_1(i)] \neq 0$ and $BF[h_2(i)] \neq 0$, because $CBF[h_1(i)] \neq 0$ and $CBF[h_2(i)] \neq 0$. Therefore, $\{CBF_C[h_1, h_2(i)] \cap BF[h_1, h_2(i)]\} \neq 0$, if $i \in C \cap S$. Please look at those positions in $BF$, where the number color is red and the number is 1. After encryption and decryption, these positions are still representations of set intersection in the array $PBF$, i.e., $j \in red \land PBF[red] = 1$, if $i \in C \cap S$ and $j \in \{h_1(i), h_2(i)\}$. Furthermore, $KBF$ is an encrypted array of $BF$ by the key $k^*$, where SP knows all elements. This array $PBF$ is an array that partially decrypts $KBF$ with the key $k^*$, where the client only knows part of the elements. In our example, $j \in \{2, 4, 10, 14\}$, if $i \in C \cap S$. Then, the client uses the array $PBF$ to obtain the set intersection $\chi$, i.e., $\{34, 56\}$, by Algorithm 4.

*5.2. Security*

The protocol consists of three main parts, i.e., key generation phase, encryption phase and decryption phase. The security analysis of the protocol will be orderly presented.

5.2.1. SP privacy

During key generation, the security of Step 1 is guaranteed by Xiao et al.'s OQKD protocol [29]. By the analysis of reference [29], a dishonest client will not receive more bits than expected, i.e., $m + q$-bit, even with more efficient measures, such as the optimal unambiguous state discrimination (USD) measurement.

During the encryption phase, SP firstly maps a private set $S = \{s_1, s_2, ..., s_w\}$ to an array $CBF = \{CBF_1, CBF_2, ..., CBF_\tau\}$ through hash functions $\{h_1, h_2, ..., h_\lambda\}$ that the client also knows. Then, SP changes $CBF = \{CBF_1, CBF_2, ..., CBF_\tau\}$ to obtain an array $BF = \{BF_1, BF_2, ..., BF_\tau\}$. That is, if a dishonest client obtains $BF$, she may obtain SP's private set $S = \{s_1, s_2, ..., s_w\}$. However, SP encrypts $BF$ by the key $k^*$, where SP knows all the bits of the key, while the client only knows the partial bits. The security of $BF$ has information-theoretic security because SP uses one-time pad encryption. During the decryption phase, the client can just decrypt the encrypted array $KBF$ to obtain partial values of $BF$ by $k^*$, where she only knows $m$-bit of the key. That is to say, the client cannot have more information about SP's private set $S$.

In a word, the protocol can protect the privacy information of SP.

5.2.2. Client Privacy

Specifically, if a dishonest SP wants to eavesdrop on the client's private key during the key generation phase, the probability that his dishonesty will be detected by his client is at least $1 - \frac{1}{2^q}$, where q is a secure parameter.

The security in Step 1 of key generation is guaranteed by Xiao et al.'s OQKD protocol [29]. Based on reference [29], a dishonest SP will introduce bit errors. That is, if SP obtains a message on the conclusiveness of the client's bits, he will lose information on the bit values that the client has recorded. Actually, it is impossible for SP to have both correct bit value and conclusiveness message of the client's measurement, i.e., position index of the correct basis. Therefore, SP cannot simultaneously obtain a bit value $k_b(j)$ that is a correct result deciphered by the client and its corresponding index $j$.

In Step 2 of key generation, the client randomly compares q bits of the key with corresponding bits announced by SP to decide whether SP is dishonest. SP cannot know which bits will be taken as the checked bits before the client declares them.

Moreover, for each checked bit, if SP does not honestly execute the protocol, he will receive an error probability of $\frac{1}{2}$ in the honesty test. Therefore, for a dishonest SP, the successful probability of completely passing the honest test is less than $\frac{1}{2^q}$.

Finally, in Step 4 of key generation, the client declares the permutation $\pi$ to SP, which is defined by two sets $\{j_1, j_2, ..., j_m\}$ and $\{p_1, p_2, ..., p_m\}$. Next, the condition probability $P(\{j_1, j_2, ..., j_m\}, \{p_1, p_2, ..., p_m\}|\pi)$ will be analyzed. Although the permutation $\pi$ is randomly selected by the client, it still must satisfy Equation (1). That is, the client announces

a random permutation $\pi$ with m fixed points, where fixed points are private, but the permutations are public. Accordingly, the number of permutations satisfies the condition $m!(\tau - m)!$.

For simplicity, suppose that $JM$ denotes two arrays $\{j_1, j_2, ..., j_m\}$ and $\{p_1, p_2, ...p_m\}$. $p(|)$ and $I(;)$ denote the conditional probability and mutual information, respectively. Then, we deduce following results:

$$P(\pi) = \frac{1}{\tau!} \tag{3}$$

$$P(\pi \mid JM) = \frac{1}{m!(\tau - m)!} \tag{4}$$

$$P(JM) = \frac{1}{C_\tau^m \cdot C_\tau^m} \tag{5}$$

$$
\begin{aligned}
I(\pi; JM) &= \log \frac{P(\pi \mid JM)}{P(\pi)} \\
&= \log \frac{\frac{1}{t!(\tau - m)!}}{\frac{1}{\tau!}} = \log \frac{\tau!}{t!(\tau - m)!}
\end{aligned}
\tag{6}
$$

$$
\begin{aligned}
I(JM) &= -\log P(JM) = -\log \frac{1}{C_\tau^m \cdot C_\tau^m} \\
&= 2 \log C_\tau^m = 2 \log \frac{\tau!}{t!(\tau - m)!}
\end{aligned}
\tag{7}
$$

$$
\begin{aligned}
I(JM \mid \pi) &= I(JM) - I(\pi; JM) \\
&= 2 \log \frac{\tau!}{t!(\tau - m)!} - \log \frac{\tau!}{t!(\tau - m)!} \\
&= \log \frac{\tau!}{t!(\tau - m)!}
\end{aligned}
\tag{8}
$$

$$I(JM \mid \pi) = -\log P(JM \mid \pi) \tag{9}$$

$$P(JM \mid \pi) = \frac{1}{\frac{\tau!}{m!(\tau - m)!}} = \frac{1}{C_\tau^m} \tag{10}$$

The probability of successfully guessing values of two arrays $\{j_1, j_2, ..., j_m\}$ and $\{p_1, p_2, ..., p_m\}$ through the public permutation $\pi$ is negligible, i.e., $\frac{1}{C_\tau^m}$.

As we know that $p(M) = \frac{1}{C_\tau^m}$, so $p(JM|\pi) = p(M)$. In other words, the probability of successfully guessing these sets $\{j_1, j_2, ..., j_m\}$ and $\{p_1, p_2, ...p_m\}$ with the public permutation $\pi$ is equal to the probability of directly guessing values of set $\{p_1, p_2, ...p_m\}$ without $\pi$. In addition, the set $\{j_1, j_2, ..., j_m\}$ is the client's private message. Therefore, it is difficult for the SP to obtain the private set $\{p_1, p_2, ...p_m\}$ even if the client declares the permutation $\pi$.

In a word, the honest test (i.e., $q$ checked bits) ensures the honesty of SP during the key generation phase. The probability of successfully guessing the private sets by the public permutation $\pi$ is negligible, i.e., $\frac{1}{C_\tau^m}$.

Furthermore, the client does not send any information during encryption and decryption phases, so private information is not leaked. Therefore, the protocol can protect the client's private information.

### 5.3. External Security Analysis and Anonymity Analysis

In this protocol, we not only consider the three basic properties above but also consider external adversary security analysis and anonymity analysis.

### 5.3.1. External Security Analysis

During the key generation phase, the external security of Step 1 is guaranteed by Xiao et al.'s OQKD protocol [29]. Their protocol is resistant to external attacks, such as impersonation and man-in-the-middle attacks, through quantum bits for authentication (QA). Thus, our protocol can resist external attacks in the key generation phase.

Furthermore, they also use quantum bits to generate a key $K$, which is shared by SP and the client.

In the Step 7 of the encryption phase, even if a malicious adversary impersonates the client, she cannot obtain SP's private set $S = \{s_1, s_2, ..., s_w\}$ by the encrypted quantum sequence $\{|a_1\rangle, |t_1\rangle, |a_2\rangle, |t_2\rangle, ..., |a_\tau\rangle, |t_\tau\rangle\}$. First, the quantum sequence is obtained by encrypting the array $KBF$ with the key $K$. The adversary cannot obtain values of $K$, which is only known by the client and SP. Secondly, the adversary also cannot know the values of the array $KBF$, where the security is information-theoretic security. Therefore, even if the adversary pretends to be the client to obtain the quantum sequence $\{|a_1\rangle, |t_1\rangle, |a_2\rangle, |t_2\rangle, ..., |a_\tau\rangle, |t_\tau\rangle\}$, he cannot obtain SP's private information.

Furthermore, a malicious adversary may apply man-in-the-middle attack in Step 7 of encryption and Step 9 of decryption phases. She first intercepts the quantum sequence sent by SP and then sends fake information to the client so that the client decrypts fake information. However, the client verifies the correctness of the transmitted information. Once any bit is wrong, the client will think there is an external adversary or SP is dishonest. The adversary cannot obtain values of the key $K$, so fake information cannot pass verification. Therefore, in the encryption and decryption phases, our protocol can resist impersonation and man-in-the-middle attacks.

In a word, the protocol can resist external attacks, such as impersonation and man-in-the-middle attacks.

### 5.3.2. Anonymity Analysis

In the key generation phase, the anonymity analysis of Step 1 is guaranteed by Xiao et al.'s OQKD protocol [29]. They send quantum sequences through CA.

During the encryption phase, SP only sends quantum sequences $\{a_1, t_1, a_2, t_2, ..., a_\tau, t_\tau\}$ to clients that SP already knows in step 1 of the key generation phase. However, in decryption phase, the client cannot directly determine whether the quantum sequence $\{a_1, t_1, a_2, t_2, ..., a_\tau, t_\tau\}$ is sent from the actual SP, even if quantum information indeed comes from SP. Because the client cannot determine the source of the quantum sequence. Therefore, the protocol provides an authentication function. That is, if the quantum sequence $\{a_1, t_1, a_2, t_2, ..., a_\tau, t_\tau\}$ passes authentication, the sequence is indeed sent by SP. After the quantum sequence $\{a_1, t_1, a_2, t_2, ..., a_\tau, t_\tau\}$ are authenticated, the client can obtain the actual encrypted array $KBF$ from SP.

Therefore, the protocol can guarantee the anonymity of the communicating parties.

### 5.4. Performance

In the key generation of the protocol, it uses single photons as quantum resources. There are no complicated quantum operators except projective measurements of single photons and simple single-bit operators. In encryption and decryption, the protocol only uses simple single-bit operators and projective measurements of single photons; thus, it is easy to implement this protocol in a real-life setting.

Next, we will consider the role of protocol in updatable databases. In encryption and decryption, counting Bloom filters are employed to reduce communication overhead and accommodate dynamic databases. Counting Bloom filters are employed to handle the updated data from Algorithms 2 and 3. With the increase in data, we only need to change corresponding values in the count Bloom filter according to updatable values, instead of creating a completely new Bloom filter at each modification. At the same time, the size $\tau$ of the count Bloom filter will not be changed when updating the database on a small scale.

Instead, the protocol only increases the size of the counting Bloom filter to reduce the false rate after that data increases to a certain threshold.

With the size $\tau$ of the count Bloom filter remaining the same, if the client needs more key bits due to the increase in data, the client only needs to request insufficient key bits from SP, not all bits of the key. For example, the client and SP had the key $k_1^*$ of the $\tau$ length, where the client only knew $k$-bit values of the key, while SP knew all bits of the key. Now, the client has the size $l(l > k)$ of position indexes of non-zero items in the count Bloom filter. Then, the client only needs to obtain a new key $k_2^*$ of the $p(p <= \tau)$ length from SP, where the client only knows $(k - l)$-bit. Later, the client combines the key $k_1^*$ and $k_2^*$ to form a new key $k_3^*$ of $\tau$ length after applying the permutation $pl$, where the client knows $l$-bit of $k_3^*$. The effect of $pl$ is similar to the effect of $\pi$. Then, the client announces the permutation $pl$ to SP. In this way, we can reduce the communication overhead of keys and the cost of preparing them. Of course, we consider the semi-honesty model, where the client should not deceive SP. Thus, the protocol can significantly reduce computation and storage overhead.

From Table 5, we can see a comparative summary of existing quantum private set intersection (QPSI) protocols. The communication complexity of our protocol is $O(\tau)$-qubit. The transmitted qubits of the OQKD protocol [29] in key generation are $\kappa(\tau + q) + z$ qubits, where $z$ is the number of the qubits for authentication, $\kappa$ is a security parameter and $\kappa \approx log\sqrt{(\tau + q)}$. Then, in the encryption phase, SP only transmits $2\tau$-qubit to the client. Therefore, the communication complexity (qubit) of our protocol depends on the communication complexity (qubit) of OQKD protocol, i.e., $O(\tau)$, because $\tau \gg q$ in $O(\kappa(\tau + q) + z)$. The client needs a single-photon measurements of the $\kappa(\tau + q) + z$-qubit in the key generation phase. CA only needs to change the quantum state of the $z$-bit in the key generation. Therefore, the computation complexity of the key is $O(\tau)$. SP needs to generate $2\tau$-qubit while performing quantum transformations on them in the encryption phase. The client needs single-photon measurements of $2\tau$-qubit in the decryption phase. Therefore, the computation complexity of transmitted messages is $O(\tau)$. The computation complexity of this protocol is $O(\tau)$.

**Table 5.** Comparison summary.

| Protocol | Ours | [24] | [22] | [18] | [19] | [20] |
|---|---|---|---|---|---|---|
| Quantum resource | single photons | single photons | single photons | multi-particle entangled states | multi-particle entangled states | multi-particle entangled states |
| Complicated oracle operators | no | no | no | yes | yes | yes |
| Dimension of the Hilbert Space | 2 | 2 | 2 | $N$ | $N$ | $N$ |
| Quantum measurements | single-photon measurements | single-photon measurements | single-photon measurements | projective measurements | projective measurements | projective measurements |
| Intersection cardinality revealed to SP | no | no | no | no | no | yes |
| Communication complexity (qubit) | $O(\tau)$ | $O(\varsigma)$ | $O(N)$ | $O(vlogN)$ | $O(vlogN)$ | $O(v+l)logN$ |
| Computation complexity (qubit) | $O(\tau)$ | $O(\varsigma)$ | $O(N)$ | $O(v)$ | $O(v)$ | $O(N+l)$ |
| Round complexity in the set intersection | 1 | 1 | 1 | 2 | 3 | 4 |
| Resistant to external attacks | yes | no | no | no | no | no |

Similarly, we analyze that the communication complexity of the protocol [24] should be $O(\varsigma)$-qubit ($N \gg \varsigma \gg q$), because the OQKD protocol [23] that they cite needs to transmit $\omega(\varsigma + q)$-qubit, where a security parameter is $\omega \approx log\sqrt{(\varsigma + q)}$. The communication complexity of the protocol [22] should be $O(N)$-qubit ($N \gg \varsigma \gg q$) because the OQKD protocol [23] that they cite needs to transmit $\omega(N + q)$-qubit, where a security parameter is $\omega \approx log\sqrt{(N + q)}$.

In addition, our protocol only needs single photons, which are easier to achieve in a real-life setting. We also have a linear communication performance $O(\tau)$, where $\tau \approx \varsigma \ll N$ and $\tau < v$ in large-scale data. We need only one round of communication during the data transfer phase, i.e., $\{|a_1\rangle, |t_1\rangle, |a_2\rangle, |t_2\rangle, ..., |a_\tau\rangle, |t_\tau\rangle\}$.

## 6. Discussion

PSI have a wide range of application environments in IoT. In this paper, a novel quantum PSI in IoT is designed with the help of OQKD, quantum authentication and count Bloom filter. We describe the correctness and security of this protocol by formal expressions. Of course, there is also some security analysis software for reference, such as AVISPA and SCYTHER. In this paper, we extend the OQKD method to PSI. In Table 6, we describe some differences between this paper and the underlying protocol.

**Table 6.** Comparison with the OQKD protocol [29].

| Protocol | Research Themes | SP (Server) Honesty Test | The Data Process | Matching Method between Keys and Data |
|---|---|---|---|---|
| Ours | private query | no | no | a shift value |
| [29] | PSI | yes | count Bloom filter | a permutation |

Below we present some limitations of the protocol and the direction of future work. Limited by the current development of quantum technology, we are not able to conduct experiments and perform practical validation of the protocols in the IoT. Although the development of quantum facilities is still immature, there already exist some programming environments capable of simulating a small number of quantum bits, e.g., HiQ quantum cloud platform, IBM quantum cloud platform. The OQKD of key generation is similar to that of quantum key distribution (QKD). As far as we know, the key rate of QKD is 14.5 b/s under experimental conditions of 75 MHz clock rate and time bin encoding [40], which is the most advanced development [41]. Quantum devices are also subject to this protocol. We hope to perform experimental validation of the protocol in the future. The OQKD protocol [29] is the first protocol that combines OQKD methods with quantum authentication, but to our knowledge, its efficiency is not optimal. In the future, we can improve the efficiency of the overall protocol with other existing OQKD protocols [42–44]. The quantum authentication method used in the overall protocol requires relatively more conditions. In the future, we will improve the authentication method with better quantum authentication protocols. In addition, we hope to combine this protocol with existing classical methods so that the protocol can contribute to the development of research in other directions [7,45]. At the same time, we would like to promote a new idea: the most likely faster implementation of OQKD or QKD as a basic building block for other research topics. We hope to combine QKD and OQKD with other technologies to create a whole new security system.

## 7. Conclusions

In this paper, we proposed a generic system model aided with ED for PSI in IoT. Then, we presented a quantum PSI protocol in IoT. Our proposed quantum PSI protocol obtained higher security and only needed the communication complexity of $O(\tau)$ qubits. The proposed protocol can not only protect the private data of two parties but also protect

identity information of two parties. The proposed protocol had an authentication function to prevent malicious adversary attacks and maintain information integrity.

**Author Contributions:** Conceptualization: B.L. and X.Z.; methodology, B.L. and X.Z.; validation: B.L., X.Z. and M.Z.; formal analysis: B.L. and M.Z.; investigation: B.L. and X.Z.; resources: M.Z.; data curation: B.L. and X.Z.; writing—original draft preparation: B.L. and X.Z.; writing—review and editing: B.L., X.Z. and R.S.; visualization: B.L. and X.Z.; supervision: M.Z. and G.Z.; project administration: B.L. and M.Z.; funding acquisition: B.L. and M.Z. All authors have read and agreed to the published version of the manuscript.

**Funding:** This work was supported by the National Natural Science Foundation of China (62002105, 62072134, U2001205, 61902116) and The Key Research and Development Program of Hubei (2021BEA163).

**Institutional Review Board Statement:** Not applicable.

**Informed Consent Statement:** Not applicable.

**Data Availability Statement:** Not applicable.

**Conflicts of Interest:** The authors declare no conflict of interest.

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
