# Peer review of "SEPSI: A Secure and Efficient Privacy-Preserving Set Intersection with Identity Authentication in IoT"

_mathematics, doi:10.3390/math10122120_

Round 1

Reviewer 1 Report

Major comments and suggestions:

1.       Please add “et al.” to the first author if a cited paper has multiple authors.

2.       The Latin phrase of “id est” should be abbreviated as “i.e.”.

3.       There are many grammar and writing style issues throughout the manuscript. However, as the lines are not numbered, it is impractical for the reviewers to point them out specifically.

4.       The manuscript should be reorganized from the current IEEE format to fit the format requirements by MDPI Mathematics.

5.       The proposed protocol is heavily based on the existing OQKD protocol by Xiao et al. Please also list this protocol in Table V to compare against the performance of the proposed one. The improvement of performance from this protocol is not clear in the current state.

6.       Practical verification of the proposed protocol, like the implementation of the protocol in an IoT system, should be provided.

7.       The manuscript stops at listing two limitations of the proposed protocol. Please try to provide some thoughts on how to address those limitations as future work.

Reviewer 2 Report

The SEPSI paper introduces a new quantum key distribution (QKD) protocol in the IoT platform. The concept formulation is interesting to know with the demand in IoT security and the progress of quantum algorithms. The authors have presented the algorithms and protocol process, which formulates the design and is claimed to be secure and efficient.

Nevertheless, the information presented can be made improved with the following suggestions:

1. The quantum device presented here needs to disclose the details with configuration.

2. The IoT device is dependent on the service provider, so it can be an overhead in communication, kindly justify your thoughts.

3. The OQKD protocol[28] is used completely or partially for random secret key bit generation is not clear. The work is quite similar with this reference.

4. The count bloom filter size limitations and advantages can be discussed more.

5. The software used for the security analysis is unknown and needs reference.

6. Performance should be disclosed with key computation time.

7. For QKD, which programming platform is used?

8. Benchmark comparison with previous protocol can be added.

9. Message and computation cost for the protocol efficiency needs to be disclosed.

10. Recent references can be added to support the need and solution. 

Round 2

Reviewer 1 Report

Most of the previous concerns were addressed. Please still perform extensive editing of English language and style. Some professional service is recommended. 

Reviewer 2 Report

Due to the lack of supporting results and complete novelty this paper is not suitable as an original article. It can be presented as a viewpoint paper.